# ON SPEEDING UP LANGUAGE MODEL EVALUATION

**Jin Peng Zhou**[*]
Cornell University

**Christian K. Belardi**[*]
Cornell University

**Ruihan Wu**[*]
University of California, San Diego

**Travis Zhang**
Cornell University

**Carla P. Gomes**
Cornell University

**Wen Sun**
Cornell University

**Kilian Q. Weinberger**
Cornell University

## ABSTRACT

Developing prompt-based methods with Large Language Models (LLMs) requires making numerous decisions, which give rise to a combinatorial search problem over hyper-parameters. This exhaustive evaluation can be time-consuming and costly. In this paper, we propose an *adaptive* approach to explore this space. We are exploiting the fact that often only few samples are needed to identify clearly superior or inferior settings, and that many evaluation tests are highly correlated. We lean on multi-armed bandits to sequentially identify the next (method, validation sample)-pair to evaluate and utilize low-rank matrix factorization to fill in missing evaluations. We carefully assess the efficacy of our approach on several competitive benchmark problems and show that it can identify the top-performing method using only 5-15% of the typical resources—resulting in 85-95% LLM cost savings. Our code is available at `https://github.com/kilian-group/banditeval`.

## 1 INTRODUCTION

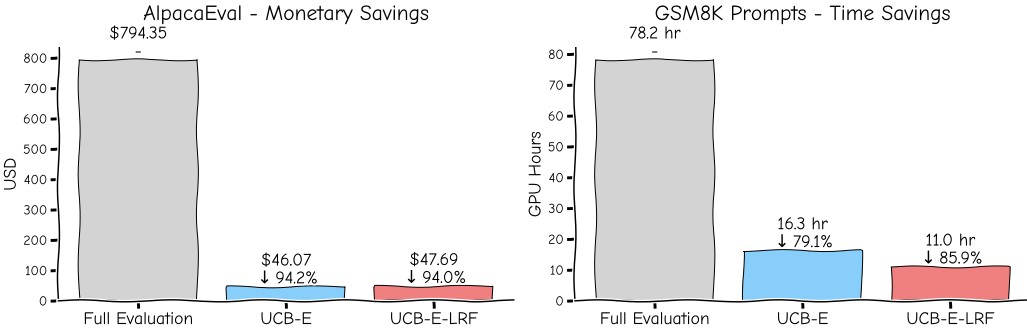

Figure 1: Cost comparison for finding the best model or prompt between our proposed algorithms (UCB-E, UCB-E-LRF) and *Full Evaluation* on two datasets. The overhead computation time for UCB-E and UCB-E-LRF is 2.4 and 142.6 seconds, respectively. See text for details.

Large language models (LLMs) have demonstrated remarkable proficiency in diverse tasks such as question answering, machine translation, and mathematical reasoning (Brown et al., 2020; Chowdhery et al., 2023; Lewkowycz et al., 2022). They are employed across numerous applications, ranging from automated customer support to content generation (Liang et al., 2024). As the development of new LLMs continues, practitioners find themselves with a plethora of options, selecting the optimal model, prompt (Kojima et al., 2022), and hyperparameters for their specific needs.

Querying an LLM is resource-intensive, and extensive evaluation requires significant investment in capital, time, and compute. Figure 1 (left) shows the estimated cost of a *Full Evaluation* of the 153 models officially included in AlpacaEval (Li et al., 2023) as of May 20, 2024 to be almost 800 USD. GPT4-turbo is the LLM judge (Zheng et al., 2024) and the monetary cost is calculated based on the

---

[*]Equal contribution. Correspondence to {`jz563, ckb73`}`@cornell.edu` and `ruw076@ucsd.edu`.

Table 1: Selected benchmarks where LLMs are state-of-the-art methods. We group them based on the task type and whether additional LLM-based / human-based scoring is needed for evaluation.

| Task Type | LLM Inference + Rule-based Scoring | LLM Inference + LLM-based Scoring |
|---|---|---|
| Natural Language Understanding | BOLD Dhamala et al. (2021) GLUE Wang et al. (2019) HellaSwag Zellers et al. (2019) SQuAD Rajpurkar et al. (2016) TriviaQA Joshi et al. (2017) WinoGrande ai2 (2019) | CNN/Dailymail Nallapati et al. (2016) Newsroom Grusky et al. (2018) XSUM Narayan et al. (2018) |
| Open-ended QA | Google NQ Kwiatkowski et al. (2019) HotpotQA Yang et al. (2018) QASPER Dasigi et al. (2021) | AlpacaEval Li et al. (2023) Chatbot Arena Chiang et al. (2024) MT-Bench Zheng et al. (2024) |
| STEM and Reasoning | APPS Hendrycks et al. (2021a) Arc Clark et al. (2018) GPQA Rein et al. (2023) GSM8K Cobbe et al. (2021) MATH Hendrycks et al. (2021c) MMLU Hendrycks et al. (2021b) PIQA Bisk et al. (2020) | – |

inference cost of only proprietary models and GPT4-turbo judging cost. Similarly, 78 Nvidia A6000 GPU hours are needed for evaluating 205 zero-shot prompts on 784 GSM8K (Cobbe et al., 2021) questions using Mistral-7B (Jiang et al., 2023), Figure 1 (right).

In this paper we investigate the familiar setting where given a *set of methods* we aim to identify the *best performing* across a *set of validation examples* — given a *fixed evaluation budget*. For example, the different methods could be LLM prompts or hyperparameter settings and the validation examples could be translation tasks or K12 math problems. The budget could be a fixed monetary sum or GPU time. We argue that this setting is extremely common among LLM applications, where the practitioner is primarily interested in identifying the single best method to deploy.

A simple baseline is to evenly split the budget for each method. However, this can be very inefficient as it would spend a lot of the budget on obvious non-performers and too little effort on identifying the very best method. Intuitively, methods that consistently underperform other methods are unlikely to end up as the best. Therefore, a more efficient budget allocation strategy is to spend less on low-performing methods and more on promising ones. Multi-armed bandit algorithms enable this dynamic allocation by actively selecting the next method-example pair to evaluate based on the results of previous evaluations.

We propose two active selection algorithms UCB-E and UCB-E-LRF. Our first algorithm is an extension of the classical UCB-E (Audibert and Bubeck, 2010) to solve the multi-armed bandit problem. It estimates the upper confidence bound (UCB) to guide the selection of the method which is paired with a randomly chosen example for the next evaluation in order to efficiently estimate the best method. This algorithm has a strong theoretical guarantee that the probability of selecting the best arm approaches 100% with an exponential convergence in the number of evaluations. Our second algorithm, UCB-E-LRF (for **L**ow-**R**ank **F**actorization), leverages the fact that in practice many validation samples are similar, and their results are *highly correlated*. In other words, if a method performs well on one sample, it will predictably also perform well on similar samples. We can therefore speculatively fill in those results and evaluate the method on a sample for which the performance is more uncertain. In our method, we capture this insight through a low-rank approximation of the evaluation matrix. By deploying this approach in addition to UCB-E, UCE-E-LRF actively selects both the method and example to evaluate, which leads to even bigger budget savings in settings with correlated validation samples.

We evaluate the efficacy of these two algorithms, in addition to common non-active baselines, in a number of practical settings. Our empirical analysis shows that our two active selection algorithms are much better than all baselines. Not surprisingly, the benefits of active selection are larger if the top methods are separated by a bigger performance gap. Notably, we observe that UCB-E works best for such easier settings; and UCB-E-LRF shines in harder settings, when the performance differences between methods are more subtle.

## 2 NOTATION AND PROBLEM FORMULATION

A typical evaluation workflow in large language model (LLM) applications involves three steps: inference, scoring, and performance aggregation.

**Inference.** Given a dataset of examples $\mathcal{X} = \{x_1, \ldots, x_n\}$, outputs are generated by specific LLM-based *methods* $\mathcal{F} = \{f_1, \ldots, f_m\}$. Here, *methods* can differ based on their LLM architecture

(architecture search), a prompt (prompt engineering (Kojima et al., 2022)), or specific decoding parameters (e.g. temperature, decoding strategies), or other hyper-parameters.

**Scoring.** The outputs from different methods are scored with a scoring function $s : \mathcal{Y} \to [0, 1]$. Without loss of generality, we assume $s(f(x)) > s(f'(x))$ indicates $f$ has better performance than $f'$ on an example $x$. The scoring function can either be computational (exact string match, BLEU (Papineni et al., 2002), ROUGE (Lin, 2004)), LLM-based (BERTScore (Zhang et al., 2019), LLM judge (Zheng et al., 2024)), or human annotators. Depending on the task and dataset format, researchers have employed different types of scoring functions. In Table 1, we classify a selected group of commonly used datasets according to the task type and scoring function applied to them. We group LLM-based and human-based scoring together since they are usually considered alternatives to each other and have been shown to have relatively high correlation (Li et al., 2023).

**Performance aggregation.** Once the underlying scoring matrix $S \in [0, 1]^{m \times n}$ is computed as $S_{ij} := s(f_i(x_j))$, the aggregate performance of method $f_i$ is its average score across all validation samples, $\mu_i = \frac{1}{n} \sum_{j=1}^{n} S_{ij}$. Alternative aggregations are possible but not considered in this work.

Both inference and scoring often rely heavily on LLM-based scoring through black-box APIs (Lambert et al., 2024)—inducing large costs and resource consumption. However, in many practical scenarios, we are only interested in identifying the best method among all methods. For example, for prompt engineering and hyperparameter tuning, knowing which method (prompt / configuration) is the best is usually sufficient for the next round of iteration. Identifying exactly which method is ranked 84 out of 100 is irrelevant and a waste of resources. We therefore claim that evaluating *all* methods on *all* examples of a dataset is excessive for this purpose, which motivates the research question of how to identify the best method $i^* = \arg\max_i \mu_i$ with limited evaluations.

Formally, given a finite evaluation budget $T$, we want an algorithm $\mathcal{A}(T, \mathcal{F}, \mathcal{X})$ that maximizes $\mathbb{P}_{\mathcal{A}}(\mathcal{A}(T, \mathcal{F}, \mathcal{X}) = i^*)$, the probability of returning the best method $i^*$ within budget $T$. We introduce the following additional notation to allow for a clear description of the algorithms. Let $S^{\mathrm{obs}} \in ([0, 1] \cup \{?\})^{m \times n}$ denote the partially observed scoring matrix, and $O \in \{0, 1\}^{m \times n}$ denote the observation matrix where $O_{ij}$ indicates whether the method-example pair $(f_i, x_j)$ has been observed. Finally, let us denote our estimate of $S$ by $\hat{S}$. For ease of reference, We also list the notations used throughout the paper in Appendix E Table 4.

## 3 ALGORITHMS

One simple baseline for designing the evaluation algorithm $\mathcal{A}$ is: uniformly sample and evaluate $T$ examples from $\mathcal{X}$, for each method $f_i \in \mathcal{F}$, estimate the performance $\hat{\mu}_i = \frac{1}{\sum_{j=1}^{n} O_{ij}} \sum_{j=1}^{n} O_{ij} \cdot S^{\mathrm{obs}}_{ij}$, and pick the method $f_{\hat{i}^*}$ with the highest estimated mean $\hat{\mu}_{\hat{i}^*}$ as the prediction for the best method. This baseline ignores any information gathered in the process of sequentially evaluating examples, and, as expected, can be very inefficient. We will see in the experiments that for some datasets, this baseline needs to evaluate at least $90\%$ of all method-example pairs to predict the best method correctly with high probability. In the following section, we will introduce two adaptive algorithms that actively select the next method-example pair to evaluate based on previous results; Figure 2 illustrates this idea on a high level for prompt tuning. The methods in the figure are different prompts, and the examples are simple math problems.

### 3.1 ALGORITHM 1 – UCB-E

The main limitation of the simple baseline is that in expectation it samples $\lfloor T/n \rfloor$ examples per method; evenly distributing its total budget across all different methods $f_i$. In order to distinguish the best method $f_{i^*}$ from other strong methods $f_i$ (i.e. $\mu_{i^*} - \mu_i$ is small), we may need more than $\lfloor T/n \rfloor$ examples, while $\leq \lfloor T/n \rfloor$ examples are sufficient to distinguish $f_{i^*}$ from obviously bad methods (i.e. $\mu_{i^*} - \mu_i$ is large).

We address this limitation by adaptively selecting example $t$ based on our previous $t - 1$ observations until our budget $T$ is expended. Our first algorithm UCB-E ($\mathcal{A}_{\mathrm{ue}}$; Algorithm 1) is a simple extension of the classic multi-arm bandit algorithm UCB-E (Audibert and Bubeck, 2010). For every method $f_i$,

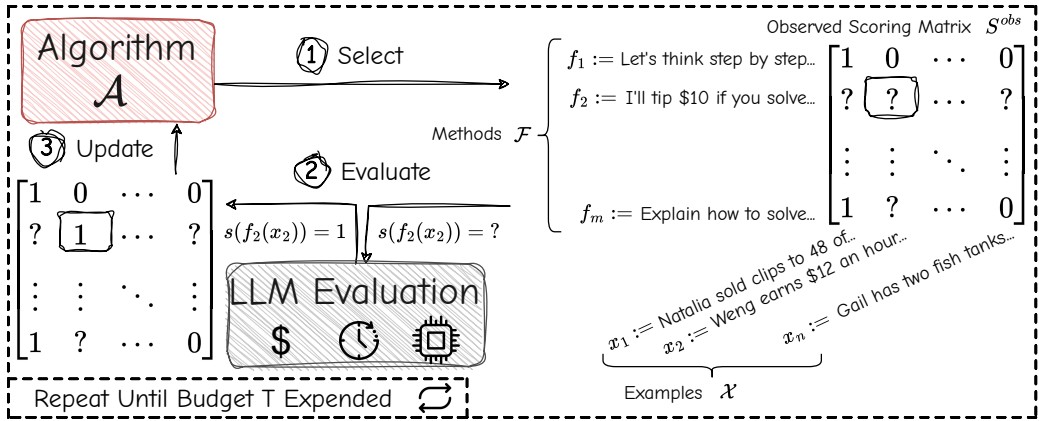

Figure 2: **Active method-example pair selection**: After LLM evaluated $t$ method-example pairs, we then call Algorithm $\mathcal{A}$ to *select* the next method-example pair. Then we *query* LLM for evaluating this pair and *fill* the scoring received from LLM into the scoring matrix. Algorithm $\mathcal{A}$ then *updates* its internal status prepared for the next method-example pair selection. This process is repeated $T$ times and, in the end, the algorithm $\mathcal{A}$ predicts the best method $f_{\hat{i}^*}$.

---

**Algorithm 1** UCB-E ($\mathcal{A}_{\mathrm{ue}}(T, \mathcal{F}, \mathcal{X}; a)$)

---

**Input:** The evaluation budget $T$, a set of methods $\mathcal{F}$, a set of examples $\mathcal{X}$, exploration parameter $a$.
**Output:** The prediction $\hat{i}^*$ for best method $i^*$.

1: The upper confidence bounds $B := \{+\infty\}^m$, the observation matrix $O := \{0\}^{m \times n}$, the observed scoring matrix $S^{\mathrm{obs}} := \{?\}^{m \times n}$.
2: **for** $t = 1, \cdots, T$ **do**
3:    **Select:** Draw $i \in \arg\max_{k \,|\, (\sum_{j=1}^n O_{kj}) \neq n} B_k$; Draw uniformly at random $j \in \{k \in [n] | O_{ik} = 0\}$.
4:    **Evaluate:** Run inference for the method-example pair $(f_i, x_j)$, score the result, and receive $s(f_i(x_j))$; $S_{ij}^{\mathrm{obs}} \leftarrow s(f_i(x_j))$.
5:    **Update:** $O_{ij} \leftarrow 1$; $B_i \leftarrow \frac{\sum_{j=1}^n O_{ij} \cdot S_{ij}^{\mathrm{obs}}}{\sum_{j=1}^n O_{ij}} + \sqrt{\frac{a}{\sum_{j=1}^n O_{ij}}}$.
6: **end for**
**Return:** $\hat{i}^* = \arg\max_i \frac{\sum_{j=1}^n O_{ij} \cdot S_{ij}^{\mathrm{obs}}}{\sum_{j=1}^n O_{ij}}$

---

at every step $t$, we estimate the upper confidence bound,

$$B_i = \frac{\sum_{j=1}^n O_{ij} \cdot S_{ij}^{\mathrm{obs}}}{\sum_{j=1}^n O_{ij}} + \sqrt{\frac{a}{\sum_{j=1}^n O_{i,j}}}.$$

Here, $a$ is a hyperparameter that controls the tradeoff between exploration and exploitation, where a larger $a$ value encourages more exploration. In the subsequent iteration, we pick the method $f_i$ with the largest upper confidence bound (ties are resolved by uniform random sampling), and then sample an example $x_j$ uniformly at random from the set of examples not yet observed, that is the set $\{j \in [n] | O_{ij} = 0\}$. We run the inference procedure for $f_i(x_j)$ and score the result, obtaining $S_{ij}^{obs} = s(f_i(x_j))$.

**Comparison with the simple baseline.** The UCB-E algorithm addresses the limitations of the simple baseline by minimizing evaluations on suboptimal methods. For a bad method (i.e. $\mu_{i^*} - \mu_i$ is large), only a few evaluations (typically proportional to $\frac{1}{(\mu_{i^*} - \mu_i)^2}$) are needed to ensure that its upper confidence bound will never exceed $\mu_{i^*}$. Consequently, the UCB-E algorithm will not select this method again, thus saving the evaluation budget and focusing on more promising methods.

**Theoretical results.** We state the theoretical guarantee of UCB-E in our context, which is a corollary of the theorem for UCB-E in the original multi-arm bandit setting (Audibert and Bubeck, 2010).

**Corollary 1 (Lower bound on success probability of UCB-E)** *Define* $H_1 = \sum_{i=1, i \neq i^*}^{m} \frac{1}{(\mu_i - \mu_i^*)^2}$ *and suppose* $a = \frac{25}{36} \frac{T-m}{H_1}$, $\mathbb{P}_{\mathcal{A}_{ue}}(\mathcal{A}_{ue}(T, \mathcal{F}, \mathcal{X}; a) = i^*) \geq 1 - 2Tm \exp\left(-\frac{T-m}{18H_1}\right)$.

*Comment.* $H_1$ in the corollary indicates the hardness of the problem specified by $(\mathcal{F}, \mathcal{X}, s)$. Intuitively, the more methods that have a smaller performance gap with the best method, the harder it is to identify which method is the best. We observe that datasets with smaller $H_1$ tend to have a higher chance of predicting $i^*$ correctly, given the same evaluation budget $T$.

## 3.2 ALGORITHM 2 – UCB-E WITH LOW-RANK FACTORIZATION (UCB-E-LRF)

Algorithm 1, the UCB-E algorithm, treats each method independently and samples examples for a method uniformly at random. In practice, however, methods and examples are correlated with each other and we can predict the outcome of a particular method-example pair from previous observations. Being able to accurately predict the outcome gives a better estimate of the score of a method. Additionally, instead of sampling examples uniformly at random, prioritizing examples with large prediction uncertainties reduces the overall uncertainty in the scores of each method, which provides more accurate score estimates in the subsequent steps. In this section, we propose a low-rank factorization based algorithm (UCB-E-LRF) to capture the above intuition.

**Low-rank factorization.** A natural approach to modeling these method-example interactions is through low-rank matrix factorization. Intuitively, if the method-examples are very correlated, there should exist $U \in \mathbb{R}^{m \times r}$ and $V \in \mathbb{R}^{n \times r}$ and rank $r \ll \min(m, n)$ such that $S \approx UV^\top$ (Candes and Recht, 2012; Chen et al., 2020; Cai et al., 2019). Obtaining $U$ and $V$ from the observed scoring matrix $S^{\text{obs}}$ allows us to estimate the scores of all method-example pairs from just a few scores. For instance, if the matrix $S$ were an exact rank-1 matrix, only $m + n - 1$ scores would be needed to recover the full scoring matrix $S$, which is far more efficient than exhaustively evaluating all $m \times n$ combinations. To find $U$ and $V$, we use *alternating least squares* (Hastie et al., 2015) to optimize the following problem:

$$\min_{U,V} \left\| O \odot \left( UV^T - S^{\text{obs}} \right) \right\|_F^2. \tag{1}$$

**Score estimation.** We can then combine the observed matrix $S^{\text{obs}}$ and estimated matrix $\hat{S} = UV^T$, using $O$ as a gating function, to arrive at a prediction of the score $\mu_i$ as follows:

$$\hat{\mu}_i = \frac{1}{n} \sum_{j=1}^{n} O_{ij} S_{ij}^{\text{obs}} + (1 - O_{ij}) \hat{S}_{ij}. \tag{2}$$

**Uncertainty estimation.** The score estimation gives us a better way to estimate the score of every method by exploiting the correlation between methods and examples at every step. We now discuss how we estimate prediction uncertainties with ensembles.

By fitting many factorization models, we obtain an empirical distribution of predictions for each method-example pair. Consequently, the standard deviation of the predictions can be used to quantify uncertainty. If the predictions are very similar to each other, this suggests the ensembles tend to agree with each other, and hence there is little uncertainty. In order to achieve sufficient model variance to approximate uncertainty, we uniformly at random drop some values from $S^{\text{obs}}$ when solving the factorization problem in Equation 1 for each model (Efron and Tibshirani, 1994; Zoubir and Boashash, 1998). For an ensemble of size $C$ the estimate for $\hat{S}$ becomes the following:

$$\hat{S} = \frac{1}{C} \sum_{c=1}^{C} U^{(c)} (V^{(c)})^T. \tag{3}$$

Then the uncertainty matrix $R \in \mathbb{R}^{m \times n}$ is given by:

$$R = \sqrt{\frac{1}{C} \sum_{c=1}^{C} \left[ \left(1 - O\right) \odot \left( U^{(c)}(V^{(c)})^T - \hat{S} \right) \right]^2}. \tag{4}$$

---

**Algorithm 2** UCB-E-LRF ($\mathcal{A}_{\mathrm{uel}}(T, \mathcal{F}, \mathcal{X}; \mathcal{M}, r, C, T_0, \eta)$)

---

**Input:** The evaluation budget $T$, a set of methods $\mathcal{F}$, a set of examples $\mathcal{X}$, the low-rank factorization model $\mathcal{M}$, the rank $r$, the ensemble size $C$, the warm-up budget $T_0$, the uncertainty scaling $\eta$.
**Output:** The prediction $\hat{i}^*$ for best method $i^*$.

1: Uniformly draw $T_0$ method-example pairs from $[m] \times [n]$ and get the observation matrix $O \in \{0, 1\}^{m \times n}$ and observed scoring matrix $S^{\mathrm{obs}} \in ([0, 1] \cup \{?\})^{m \times n}$ w.r.t. these $T_0$ evaluations.

2: $\hat{S}, R \leftarrow \mathcal{M}(S^{\mathrm{obs}}, O; r, C)$; $\forall f_i \in \mathcal{F}, B_i \leftarrow \frac{1}{n} \sum_{j=1}^n (O_{ij} S_{ij}^{\mathrm{obs}} + (1 - O_{ij})\hat{S}_{ij} + \eta R_{ij})$

3: **for** $t = T_0, \cdots, T$ **do**

4:     **Select:** Draw $i \in \arg\max_{k \mid (\sum_{j=1}^n O_{kj}) \neq n} B_k$; Draw $j \in \arg\max_{k \mid O_{ik}=0} R_{ik}$.

5:     **Evaluate:** Run inference for the method-example pair $(f_i, x_j)$, score the result, and receive $s(f_i(x_j))$; $S_{ij}^{\mathrm{obs}} \leftarrow s(f_i(x_j))$.

6:     **Update:** $O_{ij} \leftarrow 1$; $\hat{S}, R \leftarrow \mathcal{M}(S^{\mathrm{obs}}, O; r, C)$; $\forall f_i \in \mathcal{F}, B_i \leftarrow \frac{1}{n} \sum_{j=1}^n (O_{ij} S_{ij}^{\mathrm{obs}} + (1 - O_{ij})\hat{S}_{ij} + \eta R_{ij})$.

7: **end for**

**Return:** $\hat{i}^* = \arg\max_i \frac{1}{n} \sum_{j=1}^n (O_{ij} S_{ij}^{\mathrm{obs}} + (1 - O_{ij})\hat{S}_{ij})$

---

**Adaptive selection.** Bringing score estimation and uncertainty estimation together, we propose our second algorithm UCB-E-LRF ($\mathcal{A}_{\mathrm{uel}}$; Algorithm 2), an extension of Algorithm 1 leveraging the low-rank factorization model $\mathcal{M}$. The high level structure of the algorithms are very similar to each other. Specifically, we estimate the upper confidence bound (UCB) for each method based on a linear combination of score and uncertainty estimation, select the method with the greatest UCB, and evaluate the method on an example that has the highest uncertainty. Lastly, we introduce a warm-up budget $T_0$. This is because $\mathcal{M}$ requires a minimum number of initial observations in order to estimate the low-rank representation accurately. Therefore, before the active selection phase (line 3-9 in Algorithm 2), we have a warm-up phase (line 1-2 in Algorithm 2), where $T_0$ method-example pairs are sample uniformly at random to evaluate before moving on to the "active" phase.

## 4 EXPERIMENTS

### 4.1 DATASETS

Table 2: Statistics of each dataset used in our experiments. The examples $\mathcal{X}$ in each dataset correspond to the questions from their respective evaluation benchmarks. $H_1$ as defined in Section 3.1, reflects the difficulty of identifying the best method, with larger values indicating more challenging settings. More dataset details can be found in Appendix C. DA stands for Drop Annotator.

| Method Set $\mathcal{F}$ | Dataset Name | Size $m \times n$ | Scoring Function $s$ | $H_1$ |
|---|---|---|---|---|
| Various LLMs | AlpacaEval | $154 \times 805$ | GPT4-turbo annotator | 966 |
| Various LLMs excluding GPT4-turbo | AlpacaEval (DA) | $153 \times 805$ | GPT4-turbo annotator | 4462 |
| Mistral-7B with different prompts | GSM8K Prompts | $205 \times 784$ | regex match with correct answer | 107445 |
| Various LLMs and sampling configurations | GSM8K Models | $122 \times 1000$ | regex match with correct answer | 20562 |
| Tulu-7B with different prompts | PIQA Prompts | $177 \times 1546$ | regex match with correct choice | 66284 |
| Various LLMs and sampling configurations | PIQA Models | $103 \times 1000$ | regex match with correct choice | 10273 |

To assess the performance of our algorithms under a variety of use cases, we test with three datasets AlpacaEval (Li et al., 2023), *Grade School Math 8K (GSM8K)* (Cobbe et al., 2021) and *Physical Interaction: Question Answering (PIQA)* (Bisk et al., 2020), together with different settings of the method set $\mathcal{F}$ and the scoring function $s$; Table 2 summarizes each dataset. For AlpacaEval, we design two sets of $\mathcal{F}$. The first set $\mathcal{F}$ contains all LLMs reported by Li et al. (2023). The second $\mathcal{F}$ contains the same LLMs except for the annotator model GPT-4 Turbo. The latter $\mathcal{F}$ that does not contain GPT-4 Turbo makes the learning more challenging and interesting since the annotator, GPT-4 Turbo, is a clear winner among all LLMs. For GSM8K and PIQA, we set $\mathcal{F}$ as a set of prompts to simulate prompt engineering, or a set of LLMs with various sampling configurations, to mimic model selection and hyperparameter tuning. We also observe that the datasets indeed exhibit low-rank properties as seen from ratios of singular values (Table 3) and explained variance of each principal component (Figure 6) for the data matrices. More information about these datasets is in Appendix C.

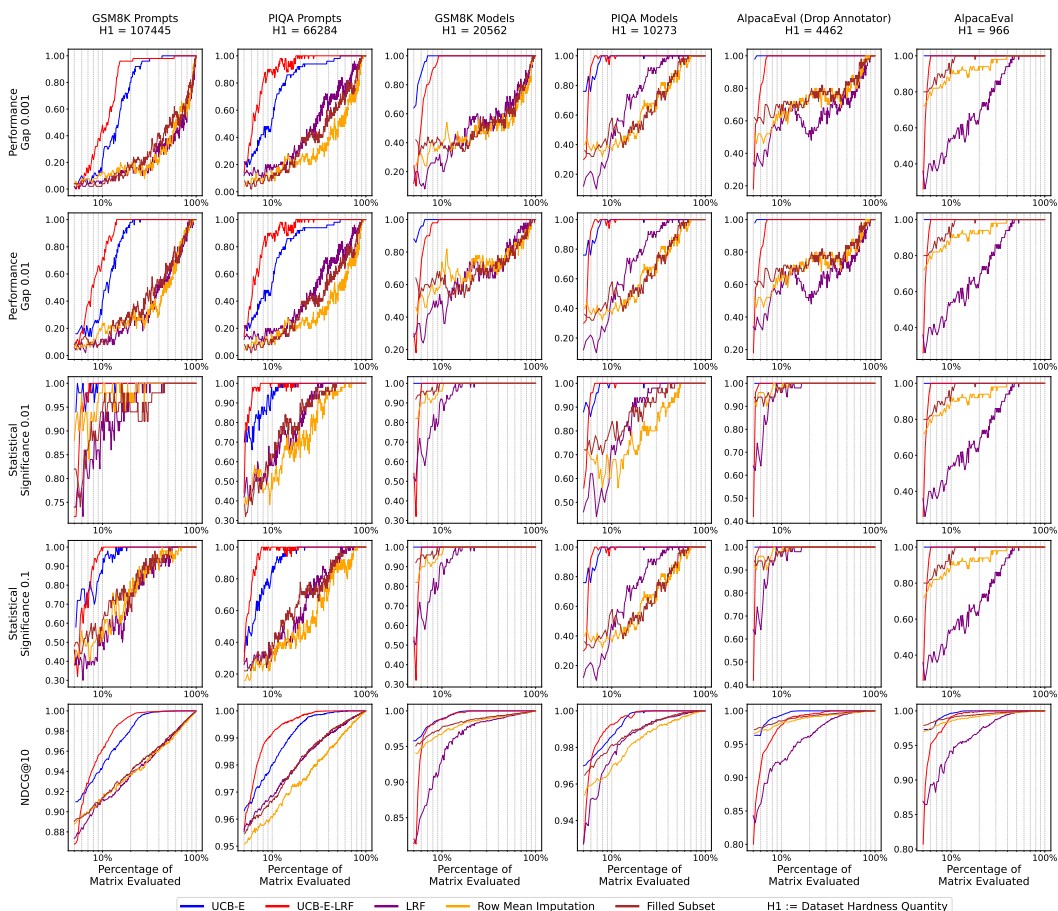

Figure 3: Comparison of algorithms on six datasets evaluated with various metrics. The vertical axes represents performance of a metric and horizontal axes represents the percentage of method-example pairs evaluated. All results are aggregated based on 50 trials with different random seeds. The datasets are ordered by decreasing $H_1$ (see Section 3.1). Larger $H_1$ values indicate settings where finding the best method is more difficult. Our proposed algorithms: UCB-E and UCB-E-LRF consistently require much less evaluation to achieve the same performance as baselines.

## 4.2 METRICS

**Top 1 Precision.** To determine if an algorithm finds the best method, we can directly check if the algorithm's prediction matches with the best method from our empirical data. However, because every method is empirically evaluated on a limited number of examples, it is possible that one method has a slightly lower average performance than another method whereas if we were to evaluate on more examples, the former would achieve a higher performance. To this end, we calculate top 1 precision by first determining a set of methods we consider equally good and checking if the predicted best method from an algorithm is a member of that set. We propose two ways to determine if a set of methods are equally good: *performance gap* and *statistical significance*. For *performance gap* $\epsilon$ top 1 precision, we consider all methods whose performance is within $\epsilon$ of the best empirical method to be equally good. For the *statistical significance* $p$ top 1 precision, we perform McNemar's statistical test (McNemar, 1947) for each method against the best empirical method. If we cannot reject the null hypothesis that the performance of one method is the same as the best empirical method up to a significance level $p$, then that method is considered equally good as the best method. We evaluate all baselines and our algorithms on two $\epsilon$ values $\{0.001, 0.01\}$, and two $p$ values $\{0.01, 0.1\}$ simulating a diverse need of precision.

**NDCG@K.** Although our focus is to identify the best method quickly, it is sometimes also desirable to generally rank the top K promising methods high. We therefore evaluate with normalized discounted

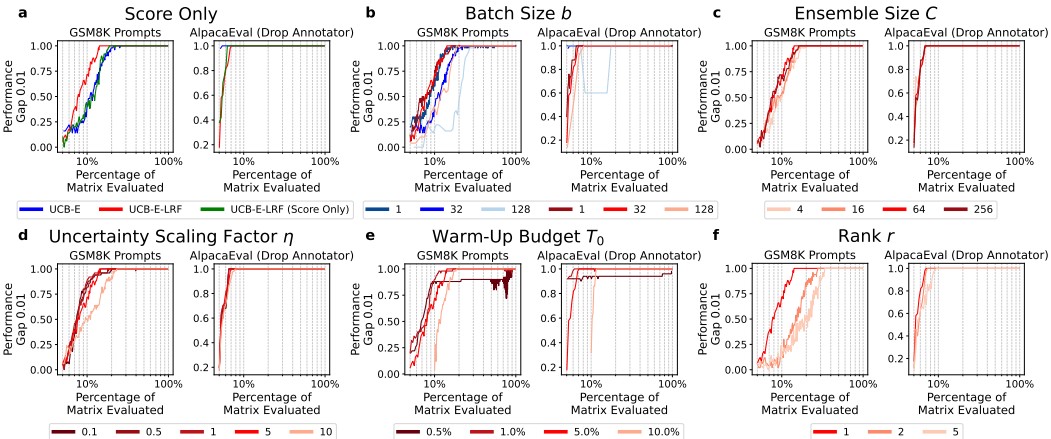

Figure 4: Ablations of the proposed algorithms and hyperparameters. UCB-E and UCB-E-LRF are denoted by blue and red lines, respectively, consistent with Figure 3.

cumulative gain (NDCG) at K. NDCG@K takes as input the top K prediction by an algorithm and the higher their true ranks are, the higher the NDCG is. We choose $K = 10$ to evaluate all algorithms.

### 4.3 SET-UP OF OUR ALGORITHMS AND BASELINES

**UCB-E:** We use $a = 1$ since this consistently yields the best performance across all datasets. This parameter is ablated in Figure 7 of Appendix D. **UCB-E-LRF:** For all datasets, we use rank $r = 1$ for low rank factorization with an ensemble size of $C = 64$. We use $5\%$ of data for warm up i.e. $T_0 = 0.05 \times m \times n$ and $\eta = 5$. **Row Mean Imputation:** We select method-example pairs uniformly at random from all pairs. The score for each method is calculated as the average of all the scores evaluated so far for that method. **Filled Subset:** Instead of randomly selecting method-example pairs, filled subset first selects an example index $j$ uniformly at random. Then all methods are evaluated on example $j$. If all methods have been evaluated, the algorithm selects a new example index from remaining ones. The score for each method is calculated as the average of all the scores evaluated so far for that method. Although row mean imputation and filled subset do not have learning component, they both produce an unbiased estimate of the real score for each method at all times. **LRF:** Similar to row mean imputation, LRF (low rank factorization) randomly selects method-example pairs to evaluate. For estimating the score of each method, LRF calculates the average of all scores for that method. That is, for evaluated examples, the score is the actual observed score, and for examples yet to be evaluated, the score is the estimated score from LRF. The details for three baselines **Row Mean Imputation**, **Filled Subset**, and **LRF** are found in Appendix D Algorithm 3, 4, and 5, respectively.

Finally, to take advantaged of the parallelism available in modern computing hardware, all algorithms and baselines are implemented with a batch size $b$. For algorithms that randomly draw an example, they instead randomly draw $b$ examples without replacement. For algorithms that draw an example with the $\arg\max$ operator, they instead draw with the $\text{Top}-b$ operator. We use $b = 32$ for all experiments unless explicitly stated otherwise.

### 4.4 MAIN RESULTS

In Figure 3, we plot the performance of baselines and our algorithms on all six datasets (columns) as the budget $T$ increases from $5\%$ to $100\%$ of the total number of method-example pairs. In each row, we evaluate the algorithms on a different metric (either top 1 precision with different $\epsilon$, $p$ or NDCG@10). Each line in the figure is the average result over 50 independent trials with different seeds. For example, an average top 1 precision of 0.9 indicates that 45 out of 50 trials predict the best method correctly at the budget level that its x-coordinate represents. In addition, we calculate $H_1$ as defined in Corollary 1 that quantifies the difficulty of finding the best method on a dataset. Intuitively, a higher $H_1$ value suggests that the method set $\mathcal{F}$ is large or there are many methods that have similar performance with the best one and distinguishing them can be challenging. In Appendix C Figure 5, we also plot the histogram to show the distribution of performance among $\mathcal{F}$ on these datasets.

**How do our algorithms compare with the baselines?** As seen from Figure 3, both UCB-E and UCB-E-LRF consistently achieve high precision and NDCG with much less budget compared to the baselines. For example, on AlpacaEval (Drop Annotator), our proposed algorithms can reach a precision of 1 with just 8% budget whereas the baselines require 80-90%, an order of magnitude more budget needed. These results suggest that it is entirely possible to identify the best method without exhaustively evaluating all method-example pairs. They further demonstrate that the active selection algorithms (our two algorithms) are more efficient than the non-active algorithms (three baselines). Additionally, the better NDCG performance from our proposed algorithms shows that our methods can more correctly rank top-performance methods.

**How is the comparison between our two algorithms UCB-E and UCB-E-LRF?** The datasets from left to right are ranked by the hardness indicated by $H_1$. Interestingly, we find that on easier datasets such as AlpacaEval, UCB-E performs better and saves 2-3% more on budget compared to UCB-E-LRF, while on harder datasets, such as GSM8K Prompts and PIQA Prompts, UCB-E-LRF achieves higher precision faster than UCB-E, saving about 10% in absolute budget. These observations give us a hint on what algorithm to apply in practice.

## 4.5 MORE EMPIRICAL ANALYSIS

We provide more empirical analysis and the ablation results can be found in Figure 4.

**Does $H_1$ correctly reflect the hardness of a dataset in the empirical experiments?** Yes, going through Figure 3 from left to right, as $H_1$ decreases, the percentage of matrix evaluation needed to reach a precision of 1 also decreases from more than 20% to just under 5%. Moreover, the $H_1$ values seem to be related to the tasks. Prompt engineering datasets typically have higher $H_1$ possibly due to the homogeneity of prompt performance with the same LLM. In contrast, datasets that benchmark different LLMs such as AlpacaEval make it much easier to find the best performing model.

**Score Only Ablation.** To study the effect of actively selecting both the next method and next example to evaluate using uncertainty matrix $R$, we consider an ablation of UCB-E-LRF. In this variant, called UCB-E-LRF (Score Only), we modify the UCB-E algorithm by replacing its mean calculation with the mean calculated via low-rank factorization (as described in Equation 3). The upper confidence bound calculation and next example selection are the same as the original UCB-E algorithm.

The detailed algorithmic description can be found in Algorithm 6. We see that on the hard dataset GSM8K Prompts where UCB-E-LRF has a certain benefit over UCB-E, there is a significant gap between UCB-E-LRF and UCB-E-LRF (Score Only). This means that the component of uncertainty matrix $R$ in UCB-E-LRF is crucial to contributing the benefit over UBC-E.

**Batch Size $b$ Ablation.** Intuitively, a smaller batch size is more flexible and can have more fine grained selection. We experiment with $b \in \{1, 32, 128\}$, and as shown in the plot, the smallest batch size gives the best performance, especially on harder datasets whereas a batch size of 128 significantly degrades the performance. However, it incurs more overall computational cost due to fitting low rank factorization more often than a smaller batch size; for example when $b = 2$, it takes almost 16 times as much time as $b = 32$ to achieve 1.0 precision, which is ineffective.

**Ensemble Size $C$ Ablation.** We vary the ensemble size $C \in \{4, 16, 64, 256\}$ and find that very small ensemble size gives a suboptimal performance on hard datasets. There is almost no performance difference on easy datasets like AlpacaEval (Drop Annotator). The performance is robust to different $C$ as long as it is larger than 64.

**Uncertainty Scaling Factor $\eta$ Ablation.** We experiment with different uncertainty scaling values $\{0.1, 0.5, 1, 5, 10\}$. In Figure 4d, it can be seen that a certain range of $\eta$, from 0.1 to 5, gives similar performance on both two datasets. This demonstrates the robustness of selecting $\eta$.

**Warm-up Budget $T_0$ Ablation.** Our algorithm UCB-E-LRF by default randomly evaluates 5% of the method-example pairs in the data matrix before selecting actively. We analyze the effect of varying the budget $T_0$ among $\{0.5, 1, 5, 10\}\%$ on the algorithm performance on the two datasets. A very small warm-up budget with 0.5% of data can achieve decent precision initially, but fall behind compared to larger warm-up budget as more data are evaluated. In contrast, a very large warm-up budget of 10% delays the active selection algorithm too much and also achieves suboptimal performance. We therefore use 5% as a generally strong starting point. On AlpacaEval, it is possible to achieve the same performance with an even smaller warm-up budget, suggesting more savings is possible.

**Rank $r$ Ablation.** Hyperparameter $r$ adjusts the bias-variance trade-off. Empirically we experiment with $r \in \{1, 2, 5\}$ and find that $r = 1$ is consistently better than larger value counterparts. The results can be explained by the fact that a larger $r$ requires more evaluated data in order to prevent overfitting, which might not have an advantage in the limited budget setting. Therefore, we recommend using $r = 1$ for all datasets.

## 5 RELATED WORK

**Best-arm identification in multi-arm bandits.** The goal of best-arm identification (Bubeck et al., 2009; Audibert and Bubeck, 2010) is to find the arm with the highest reward by pulling these arms and getting feedback. By making an analogy, in our problem method $f_i$ is the arm and the score $\mu_i$ of $f_i$ is the reward. There are two ways to define the best-arm identification problem: fixed budget and fixed confidence. In the fixed budget setting, the budget for the arm pulls is fixed and the algorithm is designed for better chances to identify the correct best arm – our problem defined in this paper has a similar evaluation budget. UCB-E (Audibert and Bubeck, 2010) and Successive Elimination (Audibert and Bubeck, 2010) are two pioneering algorithms proposed for this setting, followed by a line of improvement (Honda and Takemura, 2011; Kaufmann et al., 2012; Cappé et al., 2013; Kaufmann et al., 2016). Qin (2022) states the optimal fixed budget best-arm identification as an open problem. In the setting of fixed confidence, the algorithms (Kalyanakrishnan et al., 2012; Kaufmann and Kalyanakrishnan, 2013; Jamieson et al., 2014) work towards fewer number of arms to guarantee the given confidence of getting the best arm. Garivier and Kaufmann (2016) gives an optimal algorithm in terms of the minimum of arms to pull. Another extension beyond the setting of fixed budget or confidence is the PAC learning framework, where the target is to maximize the chance of getting an *mostly* best arm, with a tolerance of $\varepsilon$ gap to the highest reward (Karnin et al., 2013; Jamieson et al., 2014; Chaudhuri and Kalyanakrishnan, 2019).

**LLM evaluation with multi-arm bandits.** Concurrently, Shi et al. (2024) explore using multi-arm bandit for prompt optimization in a limited budget setting. Although there are similarities in the approach, it is fair to say that their work differs substantially in focus. In contrast, our main goal is to speed up model evaluation rather than prompt optimization. Further, we also propose an algorithm that leverages low-rank factorization for better best-arm identification performance. We discuss additional related work on LLM performance evaluation functions and benchmarks in Appendix A, which are relevant but orthogonal to our problem.

## 6 DISCUSSION AND CONCLUSION

In this work we focus on identifying the top-performer with a limited budget as it is a very common problem; covering a wide range of applications particularly for practitioners who simply want to find the best prompt or model for deployment purposes. One interesting direction for future work is recovering a full or top K ranking of performers, as it provides more fine-grained information that is necessary for specific use cases such as iterating on LLM design, and tracking LLM ranking over time. Of course the opportunity for resource savings is less than the top-performer setting. In Figure 3, we evaluate our algorithms on NDCG@K as preliminary evidence supporting the possibility of resource saving recovering a top K ranking. We leave a more thorough study as future work.

A limitation of our proposed algorithms is that we assume there is a two-dimensional fixed-size scoring matrix for active method-example selection. In some real-world applications, new rows or columns can be gradually incorporated, making the matrix size dynamic. In other scenarios such as Chatbot Arena, instead of being able to decide what examples to select, we can only select a pair of models to compare for a user-specified example. We leave studying how to efficiently select the best method in these settings as another direction for future work.

In conclusion, we tackle the challenge of resource-constrained evaluation, where assessing method-example pairs is costly in terms of money, compute, and time. Our goal is to identify the best method with high probability while staying within budget. We propose two sequential decision-making algorithms: UCB-E, inspired by the classic multi-arm bandit approach, which provides a theoretical guarantee on success probability, and UCB-E-LRF, which extends UCB-E by leveraging low-rank factorization to approximate the scoring matrix. Experiments show both outperform random sampling, and we identify conditions where UCB-E or UCB-E-LRF excels.

# 7 REPRODUCIBILITY STATEMENT

Our code is available at `https://github.com/kilian-group/banditeval`.

ACKNOWLEDGEMENT

We thank Justin Lovelace, Varsha Kishore, and Katie Luo for their helpful discussion and feedback on the paper draft. JPZ is supported by a grant from the Natural Sciences and Engineering Research Council of Canada (NSERC) (567916). CKB and CPG are supported by Schmidt Sciences programs, an AI2050 Senior Fellowship and two Eric and Wendy Schmidt AI in Science Postdoctoral Fellowships; the National Science Foundation (NSF) including the NSF Research Traineeship program; the National Institute of Food and Agriculture; and the Air Force Office of Scientific Research. RW is supported by grants from NSF (CIF-2402817, CNS-1804829), SaTC-2241100, CCF-2217058, ARO-MURI (W911NF2110317), and ONR under N00014-24-1-2304. WS is supported by NSF IIS-2154711, NSF CAREER 2339395 and DARPA LANCER: LeArning Network CybERagents. This research is also supported by grants from the NSF (IIS-2107161, and IIS-1724282, HDR-2118310), the Cornell Center for Materials Research with funding from the NSF MRSEC program (DMR-1719875), DARPA, arXiv, LinkedIn, and the New York Presbyterian Hospital.

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

## A  FURTHER DISCUSSION ON RELATED WORK

**Low-rank factorization for (noisy) matrix completion.** As shown in the objective function of Equation 1, low-rank factorization is a non-convex optimization problem. Therefore a line of work focus on how to solve this non-convex optimization (Candès and Tao, 2010; Candes and Recht, 2012; Chi et al., 2019), while another line of work (Chen et al., 2020; Cai et al., 2019) study the approximation error between the estimated low-rank matrix and the target matrix in terms of $p$ when assuming the observations are i.i.d. sampled with a pre-assumed chance $p$ and the additive noise to each observation is i.i.d. Gaussian noise.

**LLM performance evaluation functions.** Tremendous effort has been devoted to developing effective evaluation functions to assess the quality of open-ended generations from language models. Early work in this direction such as BLEU (Papineni et al., 2002) and ROUGE (Lin, 2004) are rule-based that use lexical overlaps to quantify similarity between a generated response and reference. However, lexical overlaps may not align well with the underlying semantics of the text. The shortcomings of these rule-based evaluation functions motivated a line of work studying using language models (Zhang et al., 2019; Sellam et al., 2020; Yuan et al., 2021) to evaluate generations. A seminal work in this area is BERTScore which uses embeddings from a BERT model (Devlin et al., 2018) to compute similarity. More recently, LLM-as-a-Judge (Zheng et al., 2024) proposes using instruction-tuned large language models to evaluate generations. Zheng et al. (2024) finds that costly proprietary models such as GPT4 have high agreement rate with human.

**LLM performance evaluation benchmarks.** Diverse benchmarks have been developed to evaluate LLM performance across various domains including natural language understanding on translations and sentiment analysis (Bojar et al., 2016; Sahayak et al., 2015; Singh et al., 2024), mathematical and common sense reasoning (Cobbe et al., 2021; Hendrycks et al., 2021c; Bisk et al., 2020), text retrieval and question answering (Yang et al., 2015; Jin et al., 2019; Dell et al., 2023). We refer readers to Chang et al. (2024) for a more comprehensive discussion on existing benchmarks. The commonality of these benchmarks is that a ground truth answer is typically given and once the responses from LLMs are generated, they can be evaluated fairly efficiently without incurring large amount of overhead such as money and compute. Recently, a new type of benchmark or more precisely leaderboard has been created attempting to compare various LLMs to each other. The typical setup involves providing the same prompt to two LLMs and then having their responses compared side by side with another LLM, which determines which response is considered superior and thus wins. Two notable leaderboards are AlpacaEval (Li et al., 2023) and Chatbot Arena (Chiang et al., 2024). AlpacaEval has a static prompt dataset and for each prompt and LLM, the response is compared with that of a baseline model (GPT3 or GPT4). For Chatbot Arena, the prompts are submitted by users and two random LLMs are selected to respond to the user-written prompt. For both benchmarks, to aggregate the overall performance of LLMs, average win rate or ELO score can be calculated from the pairwise comparison statistics. Since the evaluation requires using another LLM to decide the better response, the evaluation incurs additional compute and / or monetary cost than the benchmarks that evaluate models based on ground truth answers.

## B  DERIVATION OF COROLLARY 1

By mostly following the proof of the original theorem of UCB-E (Section 6.2 in Audibert and Bubeck (2010)), we only need to show that

$$\mathbb{P}\left(\left|\sum_{j=1}^{n} \frac{O_{ij} \cdot S_{ij}^{\text{obs}}}{\sum_{j=1}^{n} O_{ij}} - \frac{\sum_{j=1}^{n} S_{ij}}{n}\right| < \frac{1}{5}\sqrt{\frac{a}{\sum_{j=1}^{n} O_{ij}}}\right) \geq 1 - \exp\left(-\frac{2a}{25}\right).$$

This is equivalent to prove if there are $p$ random variables $\{X_1, \cdots, X_p\}$ sampling without replacement from the finite set $\{S_{i1}, \cdots, S_{in}\}$,

$$\mathbb{P}\left(\left|\frac{X_1 + \cdots + X_p}{p} - \frac{\sum_{j=1}^{n} S_{ij}}{n}\right| < \frac{1}{5}\sqrt{\frac{a}{p}}\right) \geq 1 - \exp\left(-\frac{2a}{25}\right).$$

This can be implied by the analogous Hoeffding inequality (Hoeffding, 1994) that works for the sum of random variables sampling without replacement.

# C  ADDITIONAL DETAILS - DATASETS

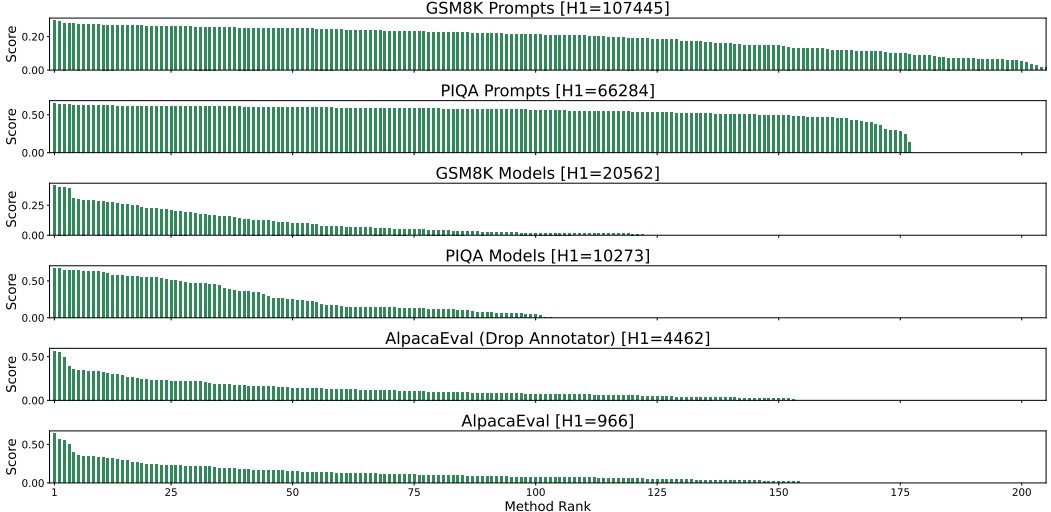

Figure 5: Bar plot of model performance on all examples in all six datasets. Examples are ordered by their rank. Dataset $H_1$ values are displayed next to dataset name. It can be seen that datasets, such as AlpacaEval, that have a large gap between the best and second best method have small $H_1$ values. The $H_1$ value indicates the difficulty of identifying the best method with smaller values indicating easier identification.

**AlpacaEval and AlpacaEval (Drop Annotator).** AlpacaEval benchmarks various models on a fixed set of 805 questions. First, for each model, the responses to these questions are collected. Then, each response is compared against that of a baseline model (GPT4-turbo) by using an LLM judge (in this case, GPT4-turbo as well). We collected the AlpacaEval dataset on May 20, 2024 from the official repository [1]. At that time, there were in total 154 models benchmarked, and hence our dataset size is $154 \times 805$. To remove the bias of favoring its own responses, we also create a derived dataset called AlpacaEval (Drop Annotator) where we drop the responses of the annotator model, which leads to a size of $153 \times 805$.

**GSM8K Prompts and PIQA Prompts.** To simulate a prompt engineering use case, we create two datasets: GSM8K Prompts and PIQA Prompts. For these datasets, we ask GPT4 to generate 205 and 177 prompts following the prompt engineering work from Yang et al. (2023). The LLM used to perform inference on the datasets are Mistra-7B Jiang et al. (2023) and Tulu-7B Wang et al. (2023) respectively. We evaluate the prompts on 784 and 1546 questions on the training set (about 10% of the size of the two training sets). For scoring, we extract the final answers from the responses generated by the LLMs and compare them with the ground truth answers (for GSM8K) or ground truth choice (for PIQA).

**GSM8K Models and PIQA Models.** Lastly, for the model selection and hyperparameter tuning use case, we similarly create two more datasets on GSM8K and PIQA. Instead of each method being a prompt, each method is now a combination of LLM with different sampling configurations. Specifically, we use 11 public available models: GPT2 [2], GPT2-Large [3], CodeLLaMA [4], Tulu-7B

---

[1]https://github.com/tatsu-lab/alpaca_eval

[2]https://huggingface.co/openai-community/gpt2

[3]https://huggingface.co/openai-community/gpt2-large

[4]https://huggingface.co/codellama/CodeLlama-7b-python-hf

[5], Tulu-2-7B [6], Gemma-7B [7], Phi2 [8], Llema-7B [9], LLaMA-2-7B [10], Mistral-7B [11] and StarCoder-7B [12]. We also have three temperature choices $\{0, 0.5, 1\}$, two maximum decoding length choices $\{128, 512\}$ and two zero-shot prompt choices (directly asking for the answer and *Let's think step by step*). The Cartesian product of all these choices give in total 132 different combinations as our methods. Depending on the dataset, some of these configurations experienced out-of-memory error on a Nvidia 3090 when we collected our data which we drop to simulate real-world scenarios. For the examples, we randomly select 1000 questions from each dataset.

Table 3: Ratio of $i^{th}$ singular value to the largest singular value of $S$ in each setting. $\sigma_i$ represents the $i^{th}$ largest singular value of $S$. The ratio $\frac{\sigma_{r+1}}{\sigma_1}$ can be interpreted as the relative reconstruction error with the best rank-$r$ approximation of $S$. For many datasets, the small numerical differences between the first row and subsequent rows suggest that using rank-1 approximation for UCB-E-LRF is appropriate.

| Singular Value Ratio | GSM8K Prompts | PIQA Prompts | GSM8K Models | PIQA Models | AlpacaEval (Drop Annotator) | AlpacaEval |
|---|---|---|---|---|---|---|
| $\sigma_2/\sigma_1$ | 0.1647 | 0.0889 | 0.3328 | 0.1972 | 0.3500 | 0.3661 |
| $\sigma_3/\sigma_1$ | 0.1588 | 0.0763 | 0.2667 | 0.1758 | 0.2153 | 0.2120 |
| $\sigma_4/\sigma_1$ | 0.1538 | 0.0739 | 0.2611 | 0.1715 | 0.1746 | 0.1754 |

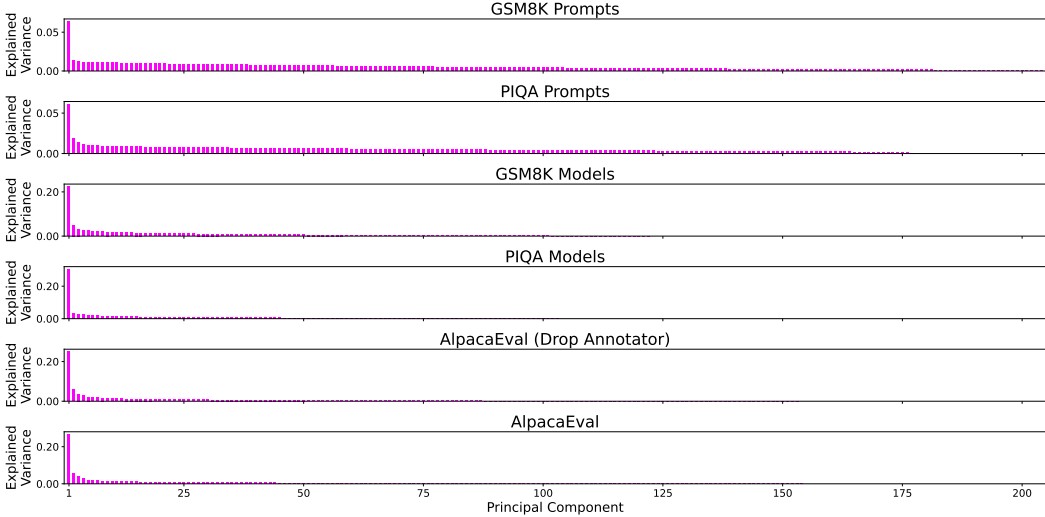

Figure 6: Bar plot of the explained variance of each principal component for all six datasets. The first principal component is much more pronounced than other principal components, suggesting the dataset matrices are generally low-rank.

## D  ADDITIONAL DETAILS - ALGORITHMS

The algorithms for Row Mean Imputation, Filled Subset, LRF and UCB-E-LRF (Score Only) are shown in Algorithm 3, 4, 5 and 6 respectively.

---

[5] https://huggingface.co/TheBloke/tulu-7B-fp16

[6] https://huggingface.co/allenai/tulu-2-7b

[7] https://huggingface.co/google/gemma-7b

[8] https://huggingface.co/microsoft/phi-2

[9] https://huggingface.co/EleutherAI/llemma_7b

[10] https://huggingface.co/meta-llama/Llama-2-7b-chat-hf

[11] https://huggingface.co/mistralai/Mistral-7B-v0.1

[12] https://huggingface.co/bigcode/starcoder2-7b

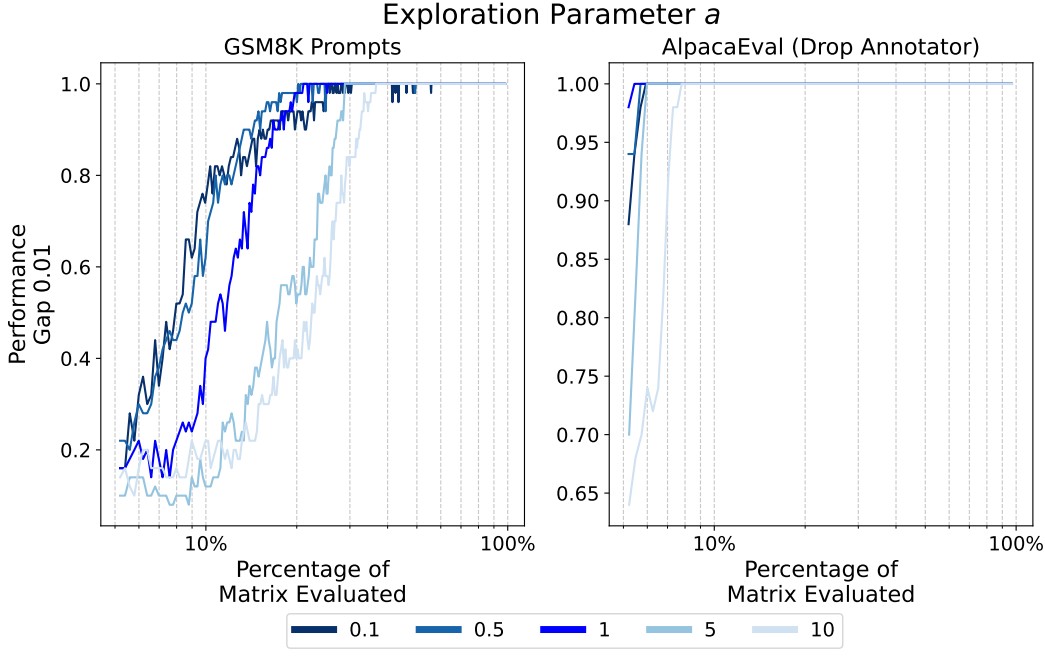

Figure 7: Ablation of the exploration parameter $a$ of UCB-E, Algorithm 1.

---

**Algorithm 3** Row Mean Imputation $(\mathcal{A}_{rmi}(T, \mathcal{F}, \mathcal{X}))$

---

**Input:** The evaluation budget $T$, a set of methods $\mathcal{F}$, a set of examples $\mathcal{X}$.

**Output:** The prediction $\hat{i}^*$ for best method $i^*$.

1: The observation matrix $O := \{0\}^{m \times n}$, the observed scoring matrix $S^{\text{obs}} := \{?\}^{m \times n}$.
2: **for** $t = 1, \cdots, T$ **do**
3:    **Select:** Draw uniformly at random $(i, j) \in \{[m] \times [n] | O_{ij} = 0\}$.
4:    **Evaluate:** Run inference for the method-example pair $(f_i, x_j)$, score the result, and receive $s(f_i(x_j))$; $S^{\text{obs}}_{ij} \leftarrow s(f_i(x_j))$.
5:    **Update:** $O_{ij} \leftarrow 1$;
6: **end for**
**Return:** $\hat{i}^* = \arg\max_i \frac{\sum_{j=1}^{n} O_{ij} \cdot S^{\text{obs}}_{ij}}{\sum_{j=1}^{n} O_{ij}}$.

---

---

**Algorithm 4** Filled Subset $(\mathcal{A}_{fs}(T, \mathcal{F}, \mathcal{X}))$

---

**Input:** The evaluation budget $T$, a set of methods $\mathcal{F}$, a set of examples $\mathcal{X}$.

**Output:** The prediction $\hat{i}^*$ for best method $i^*$.

1: The observation matrix $O := \{0\}^{m \times n}$, the observed scoring matrix $S^{\text{obs}} := \{?\}^{m \times n}$, initial random example index $j$.
2: **for** $t = 1, \cdots, T$ **do**
3:    **Select:** Draw uniformly at random $i \in \{[m] | O_{ij} = 0\}$.
4:    **Evaluate:** Run inference for the method-example pair $(f_i, x_j)$, score the result, and receive $s(f_i(x_j))$; $S^{\text{obs}}_{ij} \leftarrow s(f_i(x_j))$.
5:    **Update:** $O_{ij} \leftarrow 1$; If $\sum_{i=1}^{m} O_{ij} = m$, draw uniformly at random $j \in \{[n] | \sum_{i=1}^{m} O_{ij} = 0\}$.
6: **end for**
**Return:** $\hat{i}^* = \arg\max_i \frac{\sum_{j=1}^{n} O_{ij} \cdot S^{\text{obs}}_{ij}}{\sum_{j=1}^{n} O_{ij}}$.

---

---

**Algorithm 5** LRF ($\mathcal{A}_{lrf}(T, \mathcal{F}, \mathcal{X}; \mathcal{M}, r, C, T_0)$)

---

**Input:** The evaluation budget $T$, a set of methods $\mathcal{F}$, a set of examples $\mathcal{X}$, the low-rank factorization model $\mathcal{M}$, the rank $r$, the ensemble size $C$, the warm-up budget $T_0$.

**Output:** The prediction $\hat{i}^*$ for best method $i^*$.

1: Uniformly draw $T_0$ method-example pairs from $[m] \times [n]$ and get the observation matrix $O \in \{0, 1\}^{m \times n}$ and observed scoring matrix $S^{\mathrm{obs}} \in ([0, 1] \cup \{?\})^{m \times n}$ w.r.t. these $T_0$ evaluations.

2: **for** $t = T_0, \cdots, T$ **do**

3:     **Select:** Draw uniformly at random $(i, j) \in \{[m] \times [n] | O_{ij} = 0\}$.

4:     **Evaluate:** Run inference for the method-example pair $(f_i, x_j)$, score the result, and receive $s(f_i(x_j))$; $S^{\mathrm{obs}}_{ij} \leftarrow s(f_i(x_j))$.

5:     **Update:** $O_{ij} \leftarrow 1$; $\hat{S} \leftarrow \mathcal{M}(S^{\mathrm{obs}}, O; r, C)$.

6: **end for**

**Return:** $\hat{i}^* = \arg\max_i \frac{1}{n} \sum_{j=1}^{n} (O_{ij} S^{\mathrm{obs}}_{ij} + (1 - O_{ij}) \hat{S}_{ij})$.

---

**Algorithm 6** UCB-E-LRF (Score Only) ($\mathcal{A}_{uel-so}(T, \mathcal{F}, \mathcal{X}; \mathcal{M}, r, C, T_0, \eta)$)

---

**Input:** The evaluation budget $T$, a set of methods $\mathcal{F}$, a set of examples $\mathcal{X}$, the low-rank factorization model $\mathcal{M}$, the rank $r$, the ensemble size $C$, the warm-up budget $T_0$, the uncertainty scaling $\eta$.

**Output:** The prediction $\hat{i}^*$ for best method $i^*$.

1: Uniformly draw $T_0$ method-example pairs from $[m] \times [n]$ and get the observation matrix $O \in \{0, 1\}^{m \times n}$ and observed scoring matrix $S^{\mathrm{obs}} \in ([0, 1] \cup \{?\})^{m \times n}$ w.r.t. these $T_0$ evaluations.

2: $\hat{S}, R \leftarrow \mathcal{M}(S^{\mathrm{obs}}, O; r, C)$; $\forall f_i \in \mathcal{F}, B_i \leftarrow \frac{1}{n} \sum_{j=1}^{n} (O_{ij} S^{\mathrm{obs}}_{ij} + (1 - O_{ij}) \hat{S}_{ij} + \eta R_{ij})$

3: **for** $t = T_0, \cdots, T$ **do**

4:     **Select:** Draw $i \in \arg\max_{k \, | \, (\sum_{j=1}^{n} O_{kj}) \neq n} B_k$; Draw uniformly at random $j \in \{k \in [n] | O_{ik} = 0\}$.

5:     **Evaluate:** Run inference for the method-example pair $(f_i, x_j)$, score the result, and receive $s(f_i(x_j))$; $S^{\mathrm{obs}}_{ij} \leftarrow s(f_i(x_j))$.

6:     **Update:** $O_{ij} \leftarrow 1$; $\hat{S}, R \leftarrow \mathcal{M}(S^{\mathrm{obs}}, O; r, C)$; $\forall f_i \in \mathcal{F}, B_i \leftarrow \frac{1}{n} \sum_{j=1}^{n} (O_{ij} S^{\mathrm{obs}}_{ij} + (1 - O_{ij}) \hat{S}_{ij} + \eta R_{ij})$.

7: **end for**

**Return:** $\hat{i}^* = \arg\max_i \frac{1}{n} \sum_{j=1}^{n} (O_{ij} S^{\mathrm{obs}}_{ij} + (1 - O_{ij}) \hat{S}_{ij})$.

---

# E TABLE OF NOTATIONS

The notations used in the paper are described below.

Table 4: Notations used in the paper.

| Symbol | Description |
|---|---|
| $\mathcal{F} = \{f_1, \ldots, f_m\}$ | A list of LLM-based methods |
| $\mathcal{X} = \{x_1, \ldots, x_n\}$ | A dataset of examples |
| $s$ | A scoring function |
| $S \in ([0,1] \cup \{?\})^{m \times n}$ | A scoring matrix computed as $S_{ij} := s(f_i(x_j))$ |
| $\mu_i$ | The average score of a method $f_i$ on all examples as $\frac{1}{n} \sum_{j=1}^{n} S_{ij}$ |
| $\mu_i^*$ | The average score of the best method |
| $T$ | An evaluation budget with $T$ method-example queries |
| $\mathcal{A}(T, \mathcal{F}, \mathcal{X})$ | An algorithm that predicts the best method among $\mathcal{F}$ on $\mathcal{X}$ with $T$ query budget |
| $S^{\mathrm{obs}} \in ([0,1] \cup \{?\})^{m \times n}$ | A partially observed scoring matrix |
| $O \in \{0,1\}^{m \times n}$ | An observation matrix where 1 means the corresponding entry in $S$ has been observed |
| $B_i$ | An upper confidence bound for a method $f_i$ |
| $a$ | A hyperparameter for UCB-E algorithm that balances exploration and exploitation |
| $H_1 = \sum_{i=1, i \neq i^*}^{m} \frac{1}{(\mu_i - \mu_i^*)^2}$ | The dataset hardness quantity |
| $U \in \mathbb{R}^{m \times r}, V \in \mathbb{R}^{n \times r}$ | The low-rank factorization of $S^{\mathrm{obs}}$ such that $S^{\mathrm{obs}} \approx UV^{\top}$ |
| $\hat{S} \in \mathbb{R}^{m \times n}$ | An estimated scoring matrix $\hat{S} = UV^T$ for UCB-E-LRF |
| $\hat{\mu}_i$ | An estimated average score of a method $f_i$ for UCB-E-LRF |
| $C$ | The ensemble size for UCB-E-LRF |
| $R \in \mathbb{R}^{m \times n}$ | The uncertainty matrix for UCB-E-LRF |
| $T_0$ | The warmup budget for UCB-E-LRF |
| $r$ | The rank of low-rank factorization |
| $\mathcal{M}(S^{\mathrm{obs}}, O; r, C)$ | The low-rank factorization model that outputs $\hat{S}$ and $R$ for UCB-E-LRF |
| $\eta$ | The uncertainty scaling factor for UCB-E-LRF |

