# OpenReview forum: "On Speeding Up Language Model Evaluation"
_ICLR.cc/2025/Conference — ICLR 2025 Poster_

### Official Review · Reviewer_9sKN · 2024-10-28

**Soundness:** 2
**Presentation:** 2
**Contribution:** 2
**Rating:** 6
**Confidence:** 4

**Summary:**

This paper proposes an adaptive approach that exploits the fact that few samples can identify superior or inferior settings and many evaluations are correlated. It uses multi-armed bandits to identify the next (method, validation sample)-pair and low-rank matrix factorization to fill in missing evaluations.

**Strengths:**

(1) The studied task is interesting.
(2) The proposed method seems to be effective in acclerating the evaluation.

**Weaknesses:**

(1) The paper seems to be recycled and revised from a longer sumbisison, and many parts (especially figures or tables) are with tiny fonts, which are difficult to read.

(2) Some experimental results are difficult to understand, e.g., Table 3.

(3) Overall, I believe evaluation is quite important, and it often involves a number of influencing factors. What if there exists biases in the test datasets? How the comparison results are consistent with human evaluation, since automatic evaluation may not reflect the real capacities of LLMs?

(4) The related work part is weak, which needs more discussions on evaluation of LLMs.

**Questions:**

See the weakness

---

> ### Author Response · Authors · 2024-11-20
> **Response to Reviewer 9sKN**
>
> We thank the reviewer for the constructive feedback and pointing out the effectiveness of the proposed method. We address individual points below.
>
> > (1) The paper seems to be recycled and revised from a longer sumbisison, and many parts (especially figures or tables) are with tiny fonts, which are difficult to read.
>
> Thank you for the feedback and we apologize for the small font sizes. We have updated the PDF of our paper with larger fonts for Table 1, Table 2 and all figures. Additionally, we have added vertical grid lines for Figure 3 and 4 to make the figures more interpretable. Please let us know if there is anything we can do to further improve the presentation of our paper.
>
> > (2) Some experimental results are difficult to understand, e.g., Table 3.
>
> Thank you for the great suggestion and we have updated Table 3 in the PDF (also presented below). Both Table 3 and Figure 6 are designed to empirically verify that the scoring matrices for the datasets exhibit a low-rank structure. We use the following theorem as the foundation for Table 3:
>
> Suppose $A_r$ is any rank-$r$ matrix, then
> $$\frac{\sigma_{r+1}}{\sigma_1} = \min_{A_r} \frac{||A - A_r||_2}{||A||_2}$$
> where $\sigma_i$ denotes the $i$-th largest singular value of $A$. The LHS represents the ratio of singular values, while the RHS quantifies the reconstruction error for the best rank-$r$ approximation of $A$. Intuitively, this ratio measures how much of the total information in $A$ is captured by its top $r$ singular values.
>
> In the table below, we observe that (1) the absolute values in the first row are much smaller than 1, and (2) as $r$ increases from 2 to 3, the ratio $\frac{\sigma_{r+1}}{\sigma_1}$ decreases only slightly across all datasets. This suggests that the scoring matrices exhibit low-rank behavior (typically rank 1 or 2), as the majority of their information is captured by the top three singular values.
>
> |Dataset Name|GSM8K Prompts|PIQA Prompts|GSM8K Models|PIQA Models|AlpacaEval (Drop Annotator)|AlpacaEval|
> |-|-|-|-|-|-|-|
> |$\sigma_2/\sigma_1$|0.1647|0.0889|0.3328|0.1972|0.3500|0.3661|
> |$\sigma_3/\sigma_1$|0.1588|0.0763|0.2667|0.1758|0.2153|0.2120|
> |$\sigma_4/\sigma_1$|0.1538|0.0739|0.2611|0.1715|0.1746|0.1754|
>
> > (3) Overall, I believe evaluation is quite important, and it often involves a number of influencing factors. What if there exists biases in the test datasets? How the comparison results are consistent with human evaluation, since automatic evaluation may not reflect the real capacities of LLMs?
>
> Thank you for your insightful questions and for highlighting this important aspect of evaluation! We fully agree that biases may exist in automatic evaluation metrics, which apply to two of the six datasets we used. However, addressing these biases is orthogonal to the goal of this paper. Our primary focus is to identify the best-performing LLM or method as efficiently as possible, given any evaluation metric—whether automatic or human. Notably, our proposed algorithms are flexible and can accommodate human evaluation metrics, enabling bias reduction by simply switching from automatic to human evaluation without modifying the algorithms themselves.
>
> Furthermore, we have demonstrated the effectiveness of our algorithms on four datasets from GSM8K and PIQA, where the automatic evaluation metrics rely on regular expressions to verify whether the predicted answers match the ground truth. On these datasets, the automatic evaluation aligns closely with human evaluation. Detailed descriptions of the datasets are provided in Appendix C.
>
> > (4) The related work part is weak, which needs more discussions on evaluation of LLMs.
>
> Thank you for the great note. Due to space constraints, we have included an expanded and more detailed discussion on the evaluation of LLMs in Appendix A of our submission PDF. This section provides an overview of evaluation metrics and benchmarks of LLMs, which are relevant but orthogonal to our approach. We have also updated the main text in the related work section to reference Appendix A for a more in-depth discussion.

---

> > ### Comment · Reviewer_9sKN · 2024-11-21
> >
> > The comments clarify some of my concerns, and I slightly increase my rating score.

---

> > > ### Author Response · Authors · 2024-11-22
> > > **Response to Reviewer 9sKN**
> > >
> > > Dear Reviewer 9sKN,
> > >
> > > Thank you so much for increasing your recommendation. We are glad that our rebuttal helped address some of your concerns. If there are any other aspects we can help clarify please let us know.
> > >
> > > Regards,
> > >
> > > Authors

---

### Official Review · Reviewer_LTnJ · 2024-10-30

**Soundness:** 3
**Presentation:** 2
**Contribution:** 3
**Rating:** 6
**Confidence:** 4

**Summary:**

This paper proposes and extends the well-known multi-armed bandit (UBC) for selection of a best method/setup given a set of method (for example: an LLM, prompt, decoding parameters), a scoring function (for example: exact string matching, BLEU, ROGUE, LLM based annotator) and dataset of examples to evaluate the methods on. This extended multi-arm bandit is referred to as UBC-E. Furthermore, the paper proposes to incorporate a low-rank factorization of the observed scores to enable it to reliably interpolate missing evaluation scores. The low-rank factorization leverages the fact that method-examples are correlated with each other. The whole UBC-E-LRF conserves resources while still guaranteeing confidence that the best method/setup will be chosen.


All this is supported by theoretical proof and discussions, and furthermore shown by empirical experiments on three datasets, and various methods and setups. The UBC-E and UBC-E-LRF are compared with baselines, the top-1 precision and NDCG@10 are used as metrics.

**Strengths:**

- To the best of my knowledge, this is the first paper to be using the multi-armed bandit for LLM model/setup evaluation.
- The idea is solid, and useful. Especially, with the ever growing number of models, size of models and knobs that you can tweak to improve the performance for a specific/custom task. This framework can substantially reduce resources when practitioners have to choose a best model for their use-case.
- The algorithms are clearly outlined, making the understanding and reproduction easy.
- A big strength is that the experiments are done on multiple datasets, with varying H1. This paints a clear picture of how this framework works in different setups, and which one (UCB-E or UCB-E-LRF) to choose for which setup.
The ablations are extensive: ensemble size, uncertainty scaling factor, warm-up budget, rank, etc.

**Weaknesses:**

- In section 3.2, low-rank factorization: “Intuitively, if the method-examples are vert correlated, there should exist….” while I do agree with the intuition, it would be nice to have a citation here. At least the citation from the appendix: “Chen et al., 2020; Cai et al., 2019”.
- Even though the information is in the paper, it requires going back and forth to find it. For example, the figure captions are lacing information that is present elsewhere in the text, or not present at all. Some redundancy in the text for the sake of clarity is always welcome. I added suggestions to improve this in the Question section below.

**Questions:**

In the whole paper, only one baseline is described: uniformly sample and evaluate T examples, but three baselines are mentioned later on, and shown in the figures. What are the two other baselines? Can they be given some attention in the paper?

I have multiple small suggestions to improve the clarity and readability of the paper:
- In Table 2, the H1 value for each dataset is stated. But there is no explanation of what a higher or lower value means in the caption, or anywhere near where the table is cited. I had to refer to the Corollary 1 where it is mentioned, re-read to figure out what a higher or lower value means, to later find an explanation in section 4.4.
- In Table 2, the columns are ordered: “Dataset Name”, “Size m x n”, “Method Set”. The size is m x n, m stands for methods, n for data samples. I would either swap the “Dataset Name” and “Method Set” columns, or transform the “Size m x n” column to “Size n x m” to have a natural ordering of the columns and the order of the sizes.
- In Figure 3, there is no legend for the curve colors. In the caption of Figure 4 it is stated that the UCB-E and UBC-E-LRF are blue and red (at least for Figure 4), but there is no mention of the other curves anywhere.
- Figure 3 has the datasets ordered from highest H to lowest, and it is mentioned in 4.4 (2 pages forward) that they are ordered by hardness. There is no mention that they are ordered from hardest to easiest, and that higher H means harder and lower H means easier. It can be deducted from the whole text, but it is not immediately obvious.

---

> ### Author Response · Authors · 2024-11-20
> **Response to Reviewer LTnJ**
>
> Thank you so much for your detailed review and suggestions for us to improve the paper. We respond to the individual questions below.
>
> > In section 3.2, low-rank factorization: “Intuitively, if the method-examples are vert correlated, there should exist….” while I do agree with the intuition, it would be nice to have a citation here. At least the citation from the appendix: “Chen et al., 2020; Cai et al., 2019”.
>
> Thank you for the great suggestion and we have updated our paper PDF with added citations.
>
> > Even though the information is in the paper, it requires going back and forth to find it. For example, the figure captions are lacing information that is present elsewhere in the text, or not present at all. Some redundancy in the text for the sake of clarity is always welcome. I added suggestions to improve this in the Question section below.
>
> Thank you for the valuable paper writing suggestions! We apologize for missing captions and legends for some figures (addressed individually below). Following your and Reviewer 1wFE’s suggestions, we have also incorporated a table of notations (Appendix E Table 4) to enhance clarity and easier referencing.
>
> > In the whole paper, only one baseline is described: uniformly sample and evaluate T examples, but three baselines are mentioned later on, and shown in the figures. What are the two other baselines? Can they be given some attention in the paper?
>
> Thank you for raising this great question. In Section 4.3, we describe the three baselines: **Row Mean Imputation**, **Filled Subset**, and **LRF** with algorithmic descriptions provided in Appendix D. Row Mean Imputation and Filled Subset uniformly sample queries in two different ways, while LRF only uses low-rank factorization without any active selection.
>
> We recognize that the original paragraph formatting and the use of italic text for baseline names have made these descriptions less noticeable. To improve readability, we have updated the text in our PDF to use bold formatting for baseline names.
>
> > In Table 2, the H1 value for each dataset is stated. But there is no explanation of what a higher or lower value means in the caption, or anywhere near where the table is cited. I had to refer to the Corollary 1 where it is mentioned, re-read to figure out what a higher or lower value means, to later find an explanation in section 4.4.
>
> > Figure 3 has the datasets ordered from highest H to lowest, and it is mentioned in 4.4 (2 pages forward) that they are ordered by hardness. There is no mention that they are ordered from hardest to easiest, and that higher H means harder and lower H means easier. It can be deducted from the whole text, but it is not immediately obvious.
>
> Thank you for the suggestion. We have updated our paper PDF with a more detailed description of $H_1$ in Corollary 1, Table 2 and Figure 3. Specifically, we reference the original definition of $H_1$ and explicitly state a higher $H_1$ means a harder setting.
>
> > In Figure 3, there is no legend for the curve colors. In the caption of Figure 4 it is stated that the UCB-E and UBC-E-LRF are blue and red (at least for Figure 4), but there is no mention of the other curves anywhere.
>
> Sorry for the confusion, we sincerely apologize for missing the legend for Figure 3. We have added the legend in our revised PDF. The reviewer is also correct that the curve colors for UCB-E and UCB-E-LRF are consistent between Figure 3 and 4.
>
> > In Table 2, the columns are ordered: “Dataset Name”, “Size m x n”, “Method Set”. The size is m x n, m stands for methods, n for data samples. I would either swap the “Dataset Name” and “Method Set” columns, or transform the “Size m x n” column to “Size n x m” to have a natural ordering of the columns and the order of the sizes.
>
> Thank you for the suggestion and we have updated our Table 2 with a more natural ordering of the columns.

---

> > ### Author Response · Authors · 2024-11-24
> > **Follow up**
> >
> > Dear Reviewer LTnJ,
> >
> > We thank you for your time and feedback, and would be happy to answer any further questions you may have before the discussion period ends. Please let us know if any issues remain and/or if there are any additional clarifications we can provide.
> >
> > If you are satisfied with our rebuttal, we would appreciate it if you could reconsider your score.
> >
> > Best Regards,
> >
> > Authors

---

> > > ### Comment · Reviewer_LTnJ · 2024-11-26
> > >
> > > Thank you for your answers. Some of my concerns have been addressed - I'm happy to increase my score.

---

> > > > ### Author Response · Authors · 2024-11-27
> > > > **Response to Reviewer LTnJ**
> > > >
> > > > Dear Reviewer LTnJ,
> > > >
> > > > Thank you so much for increasing your score. We are glad that our response helped address some of your concerns. If there are any other aspects we can help clarify please let us know.
> > > >
> > > > Regards,
> > > >
> > > > Authors

---

### Official Review · Reviewer_MJvF · 2024-11-03

**Soundness:** 3
**Presentation:** 2
**Contribution:** 3
**Rating:** 6
**Confidence:** 4

**Summary:**

This paper investigates the evaluation problem of large language models (LLMs) and proposes a UCB-based evaluation method that can identify the optimal LLM strategy for a specific task with a lower budget.

**Strengths:**

The paper introduces UCB-E and its variant UCB-E-LRF.

The authors conducted extensive experiments across multiple datasets and performed repeated random seeds, which enhance the stability of the results.

**Weaknesses:**

- Some descriptions in the paper are unclear. For instance, Figure 3, which presents key experimental results, lacks a legend, making it difficult to interpret.

- Additionally, the paper does not clearly define the baseline methods used in the experiments.

- Some results also lack in-depth discussion. For example, Figure 3 shows that UCB-E and UCB-E-LRF perform inconsistently across different datasets. The authors attribute this to varying dataset difficulty; however, when comparing dataset pairs like (GSM8K Models, PIQA Models) and (GSM8K Prompts, PIQA Prompts), the conclusions are contradictory. More detailed explanation and discussion from the authors are needed here.

**Questions:**

In the experimental setup, the authors set the rank r = 1or UCB-E-LRF. Although the ablation study shows that r=1 achieves the best performance, this choice appears overly low compared to related research. The authors should provide more evidence to demonstrate that this is not due to sampling error or an inadequately small dataset.

---

> ### Author Response · Authors · 2024-11-20
> **Response to Reviewer MJvF (Part 1)**
>
> We thank the reviewer for pointing out omitted details and potential points of confusion. We address individual questions below.
>
> > Some descriptions in the paper are unclear. For instance, Figure 3, which presents key experimental results, lacks a legend, making it difficult to interpret.
>
> Apologies for the oversight and the confusion caused by the missing legend for Figure 3. We have added the legend in our revised PDF.
>
> > Additionally, the paper does not clearly define the baseline methods used in the experiments.
>
> Thank you for bringing this to our attention. In Section 4.3, we describe the three baselines: **Row Mean Imputation**, **Filled Subset**, and **LRF**. Their algorithmic details are provided in Appendix D. Row Mean Imputation and Filled Subset uniformly sample queries in two different ways, while LRF only uses low-rank factorization without any active selection.
>
> We recognize that the original paragraph formatting and the use of italic text for baseline names have made these descriptions less noticeable. To improve readability, we have updated the text in our PDF to use bold formatting for baseline names.
>
> > Some results also lack in-depth discussion. For example, Figure 3 shows that UCB-E and UCB-E-LRF perform inconsistently across different datasets. The authors attribute this to varying dataset difficulty; however, when comparing dataset pairs like (GSM8K Models, PIQA Models) and (GSM8K Prompts, PIQA Prompts), the conclusions are contradictory. More detailed explanation and discussion from the authors are needed here.
>
> Thank you for this excellent question, we have conducted additional analysis exploring this observation! While dataset difficulty, as indicated by $H_1$, is a significant factor in the performance differences between UCB-E and UCB-E-LRF, it is not the only one. We have identified that the singular value ratios of the datasets also play a critical role. Our analysis is grounded in the following theorem:
>
> Suppose $A_r$ is any rank-$r$ matrix, then
> $$\frac{\sigma_{r+1}}{\sigma_1} = \min_{A_r} \frac{||A - A_r||_2}{||A||_2}$$
> where $\sigma_i$ denotes the $i$-th largest singular value of $A$. The LHS represents the ratio of singular values, while the RHS quantifies the reconstruction error for the best rank-$r$ approximation of $A$. Below we present the singular value ratios for all datasets.
>
> |Dataset Name|GSM8K Prompts|PIQA Prompts|GSM8K Models|PIQA Models|AlpacaEval (Drop Annotator)|AlpacaEval|
> |-|-|-|-|-|-|-|
> |$\sigma_2/\sigma_1$|0.1647|0.0889|0.3328|0.1972|0.3500|0.3661|
> |$\sigma_3/\sigma_1$|0.1588|0.0763|0.2667|0.1758|0.2153|0.2120|
> |$\sigma_4/\sigma_1$|0.1538|0.0739|0.2611|0.1715|0.1746|0.1754|
>
> Intuitively, smaller singular value ratios indicate that UCB-E-LRF is better able to approximate the underlying scoring matrix, enhancing its effectiveness. In contrast, a higher $H_1$ (more difficult dataset) reflects more methods with smaller performance gaps relative to the best method, increasing the queries required for UCB-E, which does not leverage low-rank factorization.
>
> For the dataset pairs mentioned, the apparent contradictions (e.g., GSM8K Models vs. PIQA Models, and GSM8K Prompts vs. PIQA Prompts) can be reconciled by considering both the relatively high $H_1$ and the low reconstruction error (as indicated by the singular value ratios) in the table above. Datasets with smaller singular value ratios allow UCB-E-LRF to perform better due to improved low-rank approximations. Conversely, datasets with higher $H_1$ demand more evaluations for UCB-E. We will incorporate this expanded analysis into the final version of the paper to provide a more comprehensive discussion.

---

> > ### Author Response · Authors · 2024-11-20
> > **Response to Reviewer MJvF (Part 2)**
> >
> > > In the experimental setup, the authors set the rank r = 1or UCB-E-LRF. Although the ablation study shows that r=1 achieves the best performance, this choice appears overly low compared to related research. The authors should provide more evidence to demonstrate that this is not due to sampling error or an inadequately small dataset.
> >
> > Thank you for the valuable question. We believe the observation that $r = 1$ yields the best performance for UCB-E-LRF is not due to sampling error or the dataset size. Below, we provide our rationale:
> >
> > 1. **Dataset Generation and Size:**
> >
> > The datasets were designed to be both realistic and diverse. We used various temperatures (0, 0.5, 1) and models during their creation to introduce sufficient variability and randomness. Moreover, the datasets are sufficiently large. For instance, the largest dataset, PIQA Prompts, contains over 273,000 entries (177 rows by 1,546 columns). Details about the data generation process can be found in Appendix C.
> >
> > 2. **Low-Rank Structure Evidence:**
> >
> > Both the table above and Figure 6 in the PDF support the hypothesis that the scoring matrices exhibit a low-rank structure. The table demonstrates relatively low reconstruction errors for the first few rank-$r$ approximations, while Figure 6 highlights the dominant magnitude of the explained variance in the first principal component. These observations align with the ablation study results, providing additional evidence that $r = 1$ is well-suited for our datasets.
> >
> > 3. **Trade-Off Between Rank and Sampling Efficiency:**
> >
> > A higher rank $r$ increases the expressiveness of UCB-E-LRF but comes at the cost of sampling efficiency. Higher ranks require more queries to accurately learn the factorization weights, which can diminish the algorithm's efficiency in identifying the best method. The table above shows that for many datasets from $r = 1$ (first row) to $r > 1$ (other rows) provides only incremental improvements in the low-rank estimation error but introduces much higher sample complexity. Therefore, in our evaluation, we find that a very small rank, $r = 1$, is often more advantageous.

---

> > > ### Comment · Reviewer_MJvF · 2024-11-26
> > >
> > > Thank you for the authors' reply, which addressed most of my concerns. However, I have decided to maintain my current positive rating for the following reasons:
> > >
> > > - UCB-based methods have already been widely applied in other fields, limiting the overall novelty of the paper.
> > >
> > > - There are certain flaws in the paper's presentation. While the authors have stated they will revise the relevant content, the quality of the revisions cannot be guaranteed.
> > >
> > > Considering these points, along with my already positive rating, I have decided to keep my current rating.

---

> > > > ### Author Response · Authors · 2024-11-27
> > > > **Response to Reviewer MJvF**
> > > >
> > > > Dear Reviewer MJvF,
> > > >
> > > > Thank you so much for reading our response and maintaining your positive recommendation. We are glad to hear that our rebuttal addressed most of your concerns. Below we make some clarification for informational purposes.
> > > >
> > > > In addition to the classic UCB-E algorithm, the paper also presents UCB-E-LRF, an algorithm inspired by UCB-E that aims to capture and leverage the low-rank nature of scoring matrices for more efficient evaluation of LLMs. We hope this contribution broadens the range of options available to practitioners for LLM evaluations.
> > > >
> > > > We have also revised and updated several figures, tables and sections of text in the PDF. The updated text is highlighted in blue. We will make sure to incorporate the feedback from all reviewers in the final version of the paper. If there are any other aspects we can help clarify please let us know.
> > > >
> > > > Regards,
> > > >
> > > > Authors

---

### Official Review · Reviewer_1wFE · 2024-11-04

**Soundness:** 4
**Presentation:** 4
**Contribution:** 3
**Rating:** 10
**Confidence:** 3

**Summary:**

The paper proposes two active selection algorithms for evaluation based on the classical approach of estimation of the upper confidence bound. The main aim of the proposed algorithms is to identify the best performing method across a set of validation examples, given a fixed evaluation budget. The budget can be monetary cost or GPU time. The methods to evaluate can be different prompts or hyperparameter settings.

**Strengths:**

-	The proposed algorithms can be used for a variety of evaluation use cases, not limited to LLMs.
-	The paper provides enough and clear description of relevant concepts on which the proposed solution is built.
-	The paper is very well-written.
-	The proposed approaches show great money and time reduction on large evaluation datasets.
-	The approach was evaluated on variety of tasks, setups, methods, and was evaluated using a thoughtful evaluation approach.

**Weaknesses:**

-	Nothing major to report here

**Questions:**

-	Line 186: by adaptive selecting --> by adaptively selecting
-	Proof reading is needed to fix typos.
-	What does a stand for in equation in line 190?
-	I suggest some table/mapping to keep track of the different notions used and their meaning.

---

> ### Author Response · Authors · 2024-11-20
> **Response to Reviewer 1wFE**
>
> Thank you for your encouraging review and strong support! We also appreciate the reviewer for pointing out the proposed algorithms can be used for other use cases, and are not limited to LLMs. We respond to individual questions below.
>
> > I suggest some table/mapping to keep track of the different notions used and their meaning.
>
> Thank you for the great suggestion! We have incorporated a table of notations (Appendix E Table 4) in our revised PDF to enhance clarity and easier referencing. Please let us know if there is anything we can do to further improve the presentation of our paper.
>
> > Line 186: by adaptive selecting --> by adaptively selecting
>
> > Proof reading is needed to fix typos.
>
> Thank you for reading our paper carefully. We have updated the PDF with fixes to typos and grammatical errors.
>
> > What does a stand for in equation in line 190?
>
> Sorry for the confusion. $a$ is a hyperparameter that controls the tradeoff between exploration and exploitation for UCB-E, where a larger $a$ value encourages more exploration. We had added a definition for $a$ where it is first referenced in the paper.

---

### Author Response · Authors · 2024-11-20
**General Response to Reviewers**

We thank all the reviewers for their time and thoughtful feedback. We appreciate that the reviewers find our approaches **interesting** and **novel** (1wFE, LTnJ, 9sKN), and the evaluation is **extensive** (1wFE, MJvF, LTnJ) showing great **effectiveness** in cost reduction (1wFE, LTnJ, 9sKN).

We also apologize for the missing legend in Figure 3 and the less noticeable italic formatting used for baseline names in Section 4.3. Based on the insightful suggestions from all reviewers, we have revised the paper to incorporate their feedback, along with other clarifications, which are now highlighted in blue font in the updated PDF.

We respond to each reviewer's valuable critiques in our individual responses. We hope to continue this valuable discussion during the discussion period. Thank you again for your valuable input!

---

### Meta-Review · Area_Chair_kgdv · 2024-12-16

**Metareview:**

The paper presents an approach to evaluate multiple language models across a set of tasks, given a fixed evaluation budget. The idea is to expend this budget intelligently so as to quickly identify the best performing models, and not spend it on models that perform poorly. This is achieved by applying a multi-armed bandit algorithm, coupled with a low-rank estimator of the (LLM, task) score matrix.

Reviewers were unanimously supportive of the paper, finding it to be an interesting application of multi-armed bandits to a topical problem. From the AC's reading, the technical novelty may be a little restricted, but the the paper indeed executes the presented ideas well, and the work could be of broad interest to the community. The authors are encouraged to consider reporting results on a larger pool of datasets, which could further convince of the value of the proposed framework.

**Additional Comments On Reviewer Discussion:**

The initial reviews were generally positive. One reviewer had concerns around the claims of task difficulty explaining some of the performance differences between methods; specifically, the claim appeared to not be consistent with the observed results. The author response presented an alternate measure of task hardness based on the condition number, which fully explained all results. Following this, there was unanimous recommendation to accept the paper.

---

### Decision · Program_Chairs · 2025-01-22

Accept (Poster)